# A Hypothalamic Mechanism Regulates the Duration of a Migraine Attack: Insights from Microstructural and Temporal Complexity of Cortical Functional Networks Analysis

**DOI:** 10.3390/ijms232113238

**Published:** 2022-10-31

**Authors:** Camillo Porcaro, Antonio Di Renzo, Emanuele Tinelli, Vincenzo Parisi, Cherubino Di Lorenzo, Francesca Caramia, Marco Fiorelli, Giada Giuliani, Ettore Cioffi, Stefano Seri, Vittorio Di Piero, Francesco Pierelli, Giorgio Di Lorenzo, Gianluca Coppola

**Affiliations:** 1Department of Neuroscience and Padova Neuroscience Center (PNC), University of Padova, 35128 Padova, Italy; 2Institute of Cognitive Sciences and Technologies (ISTC)—National Research Council (CNR), 00185 Rome, Italy; 3Centre for Human Brain Health and School of Psychology, University of Birmingham, Birmingham B15 2TT, UK; 4IRCCS—Fondazione Bietti, 00198 Rome, Italy; 5Unit of Neuroradiology, Department of Medical and Surgical Sciences, Magna Græcia University, 88100 Catanzaro, Italy; 6Department of Medico-Surgical Sciences and Biotechnologies, Sapienza University of Rome Polo Pontino—I.C.O.T., 04100 Latina, Italy; 7Department of Human Neurosciences, Sapienza University of Rome, 00185 Rome, Italy; 8Aston Institute of Health and Neurodevelopment, College of Health and Life Sciences, Aston University, Birmingham B4 7ET, UK; 9Department of Clinical Neurophysiology, Birmingham Women’s and Children’s NHS Foundation Trust, Birmingham B4 6NH, UK; 10Laboratory of Psychophysiology and Cognitive Neuroscience, Department of Systems Medicine, University of Rome Tor Vergata, 00133 Rome, Italy; 11IRCCS—Fondazione Santa Lucia, 00179 Rome, Italy

**Keywords:** resting-state networks (RSN), fractal dimension (FD), migraine ictal, hypothalamus, diffusion tensor imaging (DTI)

## Abstract

The role of the hypothalamus and the limbic system at the onset of a migraine attack has recently received significant interest. We analyzed diffusion tensor imaging (DTI) parameters of the entire hypothalamus and its subregions in 15 patients during a spontaneous migraine attack and in 20 control subjects. We also estimated the non-linear measure resting-state functional MRI BOLD signal’s complexity using Higuchi fractal dimension (FD) and correlated DTI/fMRI findings with patients’ clinical characteristics. In comparison with healthy controls, patients had significantly altered diffusivity metrics within the hypothalamus, mainly in posterior ROIs, and higher FD values in the salience network (SN). We observed a positive correlation of the hypothalamic axial diffusivity with migraine severity and FD of SN. DTI metrics of bilateral anterior hypothalamus positively correlated with the mean attack duration. Our results show plastic structural changes in the hypothalamus related to the attacks severity and the functional connectivity of the SN involved in the multidimensional neurocognitive processing of pain. Plastic changes to the hypothalamus may play a role in modulating the duration of the attack.

## 1. Introduction

Neuroimaging studies have recently brought supporting evidence of a key role of the hypothalamus in the initiation of a migraine attack. PET studies of patients in whom the attack was experimentally induced [1] and longitudinal fMRI studies of spontaneous attacks [2,3] have shown increased hypothalamic metabolism and connectivity with the spinal trigeminal nucleus as early as 48 h before the onset of the pain phase. In the pain-free phase, hypothalamic macro-structural abnormalities and aberrant functional connectivity with frontal and temporal regions involved in the descending control of pain [4,5], with regions involved in the regulation of autonomic functions [6], and with the cerebellum [5] have been reported. Altered functional connectivity between the hypothalamus and cortical networks implicated in the pathogenesis of both interictal and ictal migraines [7] has also been observed in patients with chronic migraine [8,9]. Although there is evidence of microstructural alterations in the hypothalamus during the pain-free phase of migraine [10], data on its microstructural integrity during the attack and its relationship with cortical activity at rest are still sparse.

Current models of human brain organization indicate that anatomic architecture has a profound but not strict influence on brain function; that reflects multisynaptic interactions in complex large-scale multidimensional networks [11]. Functional brain properties cannot therefore be estimated directly from structural data but can be inferred through statistical models of varying complexity such as the ones used in our study [12]. The temporal-scale fractal properties of fMRI signal have unveiled neural network dynamics taking place in the classical three spatial dimensions [12]. Since specific anatomical organizations are associated with distinct functional patterns involving nearby and distant anatomical areas through the associated synchronous and asynchronous brain activities, this network-based approach has received recent attention in neurosciences research [13,14,15,16,17]. Using diffusion tensor imaging (DTI) data recorded in patients with migraine without aura between attacks, we have recently described abnormal proton diffusivity in the anterior and posterior hypothalamus. In the same study, we found abnormal fractal dimensionality of BOLD signals in cortical networks that support the integration of sensory, emotional, and cognitive information [18] and that correlated with the clinical presentation of migraine pain. In the present study, we investigated the microstructural integrity of the hypothalamus through DTI and the non-linear functional connectivity of the resting state network (RSN) computed from fMRI measures in a group of patients with migraine without aura in whom imaging data acquisition occurred in close temporal proximity with a spontaneous attack. We hypothesized that the hypothalamus’s microstructure and RSN connectivity would be altered during the attacks of migraine and related to migraine clinical features.

## 2. Results

Demographic characteristics of MI and HC and clinical features of MI are summarized in Table 1. No age difference emerged between MI and HC (t33 = −1.87, *p* = 0.699). We did not detect T2 or any visible white matter lesions in both patients and controls.

### 2.1. Characterization of Hypothalamic DTI

Descriptive and inferential data on DTI metrics for hypothalamic ROI are summarized in Table 2. Bilateral hypothalamic regions of interest are shown in Figure 1. The factor “group” was significant for whole hypothalamus ROI and for the posterior left and right hypothalamus ROIs. Compared to HC, patients with MI showed significantly higher MD (*p* < 0.001, Cohen’s d = 5.58), AD (*p* < 0.001, Cohen’s d = 6.01), and RD (*p* < 0.001, Cohen’s d = 5.24), with lower FA (*p* = 0.001, Cohen’s d = −2.34). In the exploratory analysis, MD (left: *p* < 0.001, Cohen’s d = 2.78; right: *p* < 0.001, Cohen’s d = 2.14), AD (left: *p* < 0.001, Cohen’s d = 2.66; right: *p* < 0.001, Cohen’s d = 2.03), and RD (left: *p* < 0.001; Cohen’s d = 2.85; right: *p* < 0.001, Cohen’s d = 2.40) were significantly higher, and the FA was lower (left: *p* = 0.01; Cohen’s d = −1.30; right: *p* = 0.001, Cohen’s d = −1.90) in the posterior hypothalamic region bilaterally. We did not detect between groups differences in the anterior hypothalamic region (Table 3).

### 2.2. fMRI Resting-State Networks

The twenty-one ICs were grouped into the following nine large-scale networks based on their spatial patterns: Auditory (IC2—AN); Cerebellum (IC1—Cb); Default Mode (DMN—IC15, IC20, IC21, IC25, IC27, IC28, and IC31); Dorsal Attention System (IC6—left DAS and IC17—right DAS); Fronto–Parietal (IC3—right FPN and IC23—Left FPN); Language (LN—IC 26); Salience Network (IC 10 and IC19); Sensorimotor (IC7, IC8, IC33); and Visual (IC11—Primary visual (PV) and IC 24—Higher visual (HVN)) networks (Figure 2).

### 2.3. Characterization of RSNs Using Higuchi’s Fractal Dimension

The general linear model for FD values showed that MI differed from HC only in the anterior part of the SN (aSN, IC10: F = 10.58, *p* = 0.001), with higher FD values of aSN in MI than HC (Cohen’s d = −1.24; Figure 3).

### 2.4. Correlation Analysis

A moderate significant negative correlation was found between the severity of migraine pain, as assessed by 0–10 VAS and FD values of IC10 (r = −0.613, *p* = 0.018, Figure 4, Upper Left Panel). When considered as a whole ROI, a moderate significant negative correlation was found between hypothalamic AD and VAS (r = −0.613, *p* = 0.015, Figure 4, Upper Right).

Considering the clinical variables, bilateral anterior hypothalamic MD, AD, and RD positively correlated with mean duration of migraine attacks (LH: MD r = 0.734, *p* = 0.002; AD r = 0.703, *p* = 0.003; RD r = 0.717, *p* = 0.003; RH: MD r = 0.722, *p* = 0.002; AD r = 0.682, *p* = 0.005; RD r = 0.691, *p* = 0.004; see Figure 5).

No other significant correlation was found between DTI metrics of the hypothalamus and FD metrics.

## 3. Discussion

Our results suggest that a migraine attack is associated with morphological and functional neural signatures. In particular, compared with healthy controls, patients showed the following: first, increased fractal dimension of areas belonging to the salience network, and, second, a general increase in FA and decrease in MD, RD, and AD diffusivity metrics of the whole hypothalamus, while the analysis of the individual hypothalamic regions revealed a prevalent involvement of the posterior hypothalamus without significant lateralization. For the correlation analysis emerged that, first, the higher the intensity of pain perceived during the attack, the higher the FD of the salience network and the AD of the whole hypothalamus, and, second, the longer the mean duration of migraine attacks, the higher the MD, RD, and AD of the bilateral anterior hypothalamic region. These findings depose complex morpho-functional changes of the migraine brain during an attack and will be interpreted in light of current pathophysiological knowledge.

### 3.1. Fractal Dimensionality

Fractal dimension is a nonlinear measure of the complexity of the cerebral hemodynamic activity. In principle, an increase or decrease in FD within a network can be related to greater or lesser flexibility and/or efficiency in information processing [19]. In this study we observed that migraine attacks are associated with increased efficiency–demand of large-scale neuronal networks.

The SN is primarily formed by the insulae and cingulate cortex and is involved in the elaboration and selection of the most relevant events to which neural resources need to be allocated [20].

It is worth noting that the salience network, which in this study showed increased FD during the attack, was previously found to be involved during the headache phase of migraine either in response to noxious stimulation [21] or at rest [22]. These results together with ours provide further evidence in favor of a reorganization of cortical functional networking during a migraine attack, likely an expression of the complex interaction between the different components that form the multidimensional nature of pain. In support of this possible interpretation, we found that high FD values within the SN correspond to a lower subjective perception of migraine pain intensity, a correlation quite comparable to that previously found in a similar group of patients between DMN-to-insula connectivity and severity of the migraine headache [23].

### 3.2. Microstructure of the Hypothalamus

Although the notion that the hypothalamus is involved in the pre-ictal period and the painful phase of migraine is receiving increasing acceptance, it is still unclear which subregion of the hypothalamus is most involved. Denuelle and colleagues [24], who first detected increased blood flow in the hypothalamus during the migraine using PET imaging, found that the region of activation was not well localized. By studying longitudinally a group of migraine patients, Schulte and colleagues [25] localized activation within the lateral anterior part of the hypothalamus before the pain phase. In contrast, other colleagues [1] described increased regional blood flow in the right posterior hypothalamus during the premonitory phase. We also detected altered hypothalamic diffusivity metrics, especially in the bilateral posterior region, while those of the anterior regions were similar to HC.

DTI technology can be useful to study not only fiber bundles but also fiber tracts in gray matter nuclei, such as the hypothalamus. However, diffusivity metrics reflect not only myelin content, but also rapid changes in membrane volume, glial cell morphology, and the number of local neuronal circuits [26]. In a previous study using the same methodology in the interictal phase, we detected significantly higher MD, AD, and RD values within the hypothalamus as a whole and in the anterior and posterior ROIs bilaterally as well, with the addition of lower FA values on the posterior ROIs [18].

The present detection of prevalent microstructural alterations of the posterior hypothalamic regions may be interpreted as suggestive of the involvement of posterior nuclei during an attack to provide a sustained vegetative response to stress, such as that associated with ongoing headache. As indirect supportive evidence, we found that the higher the axial diffusivity metric in the hypothalamus, the lower the severity of migraine.

We suggest a more general pain-induced enhancement of the hypothalamic–cortical network’s functional activity during migraine attacks, probably to modulate attention and promote emotional and visceral response to pain.

An additional novel finding of the present study is that despite the exploratory analysis, no evidence was found for group-level altered diffusive metrics in the anterior hypothalamus; measures of MD, RD, and AD in the anterior hypothalamus were associated with a longer mean duration of the attacks. We suggest that during attacks of migraine, normal-to-high functioning of anterior hypothalamic region mechanisms might participate in the process that determines the end of an attack. We hypothesize that the hypothalamus plays a complex role in a migraine attack; it not only participates in its initiation but also seems to play a role in regulating its duration and termination. This idea is not novel as it was previously hypothesized attempting to explain the role of the hypothalamus in autonomic-trigeminal cephalalgias [27]. That the clinical manifestation of migraine may be associated with a different hypothalamic activity has also been supposed by previous observation that patients who experience a worsening of the frequency of attacks during the years following the initial acquisition have reduced connectivity between the hypothalamus and the orbitofrontal cortex [5].

This study has several limitations. The first is the high variability of clinical presentation of migraine attacks which allows limited generalizability of the results, even though we observed an influence of the hypothalamus in the clinical manifestation of disease. The second is not having a sufficient number of patients to be re-investigated with the same techniques during the other phases of their migraine cycle. The third is that DTI analysis of very small ROIs in the hypothalamus is inherently vulnerable to measurement variability even though our ROIs were larger than those previously used to study the hypothalamus in migraine [2,3,6,24,25,28]. Finally, another possible limitation is the lack of correlation between morpho-functional, psychopathological, and physical variables, such as the level of anxiety and that of physical activity, which we did not check during the enrollment phase. In light of the results of the correlation analysis, it would have also been interesting to correlate hypothalamic metrics with the hours elapsed since the onset of the attack, but unfortunately, this information was not available for all patients.

Further studies designed to evaluate circadian variability of hypothalamic measures might shed light on the possible sensitivity of diffusion parameters to the different phases of the day and night. To assess reliability of our findings, future studies should be devoted to trying to reproduce this initial evidence in an independent group of patients. Moreover, it will be of interest to investigate whether the hypothalamic–cortical network also plays a crucial role in the determination of accompanying migraine symptoms, such as photophobia and allodynia, and whether taking an attack medication can relieve pain through a hypothalamic-related mechanism.

## 4. Materials and Methods

### 4.1. Participants

Among the patients consecutively seen in the headache clinic of the Sapienza University of Rome Polo Pontino, we enrolled 36 patients with episodic migraine without aura diagnosed according to the last two International Classification of Headache Disorders (ICHD), III beta and III [29] multimodal neuroimaging study. Of these patients, 15 had their MRI scans within the initial 6 h of their typical pain associated with migraine attack (MI), and their findings are reported here. All of the patients had been instructed to avoid any symptomatic treatment before completing the scan session. None of the enrolled patients had taken prophylactic therapy in the last 3 months. We also recruited 20 age-matched healthy subjects (HC) with no personal or family history of primary headaches. For all study participants, the exclusion criteria were a previous diagnosis of another neurological or psychiatric disorder or suffering from autoimmune, endocrinological, connective tissue disorders, chronic extracephalic painful or neuro-ophthalmological diseases, verified by a complete neuro-ophthalmological examination, which included assessment of visual acuity, intraocular pressure measurement, and indirect ophthalmoscopy. The scanning session took place in the early afternoon hours. We ensured that female participants were studied outside the menstrual period. This study is part of a larger study carried out from 2013 until 2019, during which participants underwent several procedures during the same experimental session. All participants provided written informed consent. This study conforms with the World Medical Association Declaration of Helsinki and was approved by the Ethics Committee of Sapienza University of Rome (RIF.CE 4839).

### 4.2. Diffusion Tensor Imaging (DTI) Data Acquisition and Analysis

MRI data were obtained on a Siemens 3T Verio scanner using a 12-channel head coil.

Diffusion tensor imaging (DTI) was acquired by using single shot echo-planar imaging (TR 12,200 ms, TE 94 ms, 72 axial slices, 2 mm thickness, isotropic voxels) [18] and using SPAIR (Spectral Attenuated Inversion Recovery) as a fat suppression technique. Images from the same participants and during the same session were obtained with diffusion gradients applied along 30 non-collinear directions, effective b values of 0 and 1000 s/mm^2^ were used.

FSL 6.0 software package (FMRIB Image Analysis Group, Oxford, England; https://fsl.fmrib.ox.ac.uk/fsl/fslwiki, accessed on 26 October 2022) was used to process image data.

The data analysis method is detailed elsewhere [18]. Images were pre-processed using the FSL toolbox DTIFIT based on a diffusion tensor model to yield FA (fractional anisotropy), MD (mean diffusivity), RD (radial diffusivity), and AD (axial diffusivity). The initial region of interest (ROI) covered the whole of the hypothalamus [30]. Thereafter, according to the coordinates provided by Boes et al. [31], for each subject, we defined 4 further ROIs covering the bilateral anterior hypothalamus—mostly with neuroendocrine function—and the bilateral posterior hypothalamus, including the wake-promoting nuclei (see Figure 1 and the resulting masks in.nii files format are available for public download at the following link: osf.io/37kxr, accessed on 26 October 2022).

Based on Baroncini et al. [30] and Boes et al. [31] methods, all ROIs were drawn using a T1-weighted template (MNI152 2 mm voxel in FSL software) by two expert neuroradiologist (E.T. & F.C). The center of gravity coordinates in MNI space for the anterior hypothalamic region are x = ±4.4; y = 1.33; z = −14.67, the posterior hypothalamic ROI coordinates are x = ±6.6; y = −7.33; z = −12.67, and those of the whole hypothalamic ROI are x = 0.136; y = 4.68; z = −11.9. The size of the hypothalamic ROIs in the 2 mm space was 6 voxels (48 mm^3^), 3 per hemisphere, equally for the anterior and posterior ROIs (Figure 1). For each participant and every hypothalamic ROI, we calculated mean FA, MD, RD, and AD values in the hypothalamus by averaging those voxels in the ROI.

### 4.3. fMRI Data Acquisition and Preprocessing

Structural anatomic scans were performed using a T1-weighted sagittal magnetization-prepared rapid gradient echo (MPRAGE) series (TR: 1900 ms, TE: 2.93 ms, 176 sagittal slices, 0.508 × 0.508 × 1 mm^3^ voxels).

Functional MRI resting state data were obtained using T2*-weighted, echo-planar imaging (TR: 3000 ms, TE: 30 ms, 40 axial slices, 3.906 × 3.906 × 3 mm^3^, 150 volumes).

Functional resting scans lasted seven minutes and 30 s, during which participants were instructed to relax, avoid motion, and keep their eyes closed, but not to fall asleep.

Data pre-processing was carried out using SPM12 software (http://www.fil.ion.ucl.ac.uk/spm/, accessed on 26 October 2022) implemented in MATLAB (version R2016 b, MathWorks, Inc., Natick, MA, USA). The first 5 volumes of each participants were discarded automatically by the MRI scanner software.

Data were realigned to the first volume to correct for head motion using a 6-parameter rigid-body process and resliced by cubic spline interpolation. Participants with a head motion greater than 2.0 mm translation or a 2.0° rotation in any direction were excluded. None of the participants was excluded based on this criterion. EPI sequence was motion-corrected by the Siemens software scanner and SPM12 toolbox. We verified groups’ subjects frame displacement, which showed similar RMS values (HC: 0.023 ± 0.009, MI: 0.024 ± 0.010) [32].

The structural (T1–MPRAGE) and functional data were co-registered for each participant dataset. The normalization procedure transformed structurally and realigned EPI images into a common stereotactic space based on Talairach and Tournoux [33], resampled by 3 mm in each direction. Finally, the spatially normalized functional images were smoothed isotropically at 8 × 8 × 8 mm.

#### fMRI Data Analysis

After data preprocessing, resting-state data of all participants as a concatenated group (HC and MI) were analyzed using spatial independent component analysis (ICA) using the infomax algorithm as implemented in the Group ICA of fMRI Toolbox (GIFT—http://trendscenter.org/software/gift/, accessed on 26 October 2022) to decompose the data into functional networks that exhibited a unique time course profile. Two data reduction steps were carried out using principal component analysis, both subject-specific and group-level steps. Firstly Subject-specific data were reduced to 50 components and subsequently reduced data were concatenated over time. Secondly, at the group level, data were reduced into 33 group independent components (ICs) using the expectation-maximization algorithm included in GIFT [34].

We propose using standard information-theoretic methods for estimating the number of components from the aggregate data set. These methods make a determination based upon the complexity or information content of the data. The number of ICs was estimated using the minimum description length (MDL) criterion [35,36]. In our specific case, 33 independent components (ICs) were indicated to be estimated. Subject-specific spatial maps and time courses were obtained using the back-reconstruction approach (GICA) [37].

From the 33 ICs, we identified the relevant RSNs by applying a previously described procedure [34]. Two experienced neuroradiologists (E.T. & F.C.) blindly reviewed the components discarding those showing spatial overlap with vascular, ventricular, and edge regions corresponding to artifacts [38]. This process resulted in twenty-one meaningful ICs that we sorted into nine functional networks, based on a manual classification blindly performed by E.T. & F.C [38,39] and on spatial correlation calculated between IC and the networks’ template achieved from GIFT (Appendix A). The functional networks were arranged in Figure 2:

Auditory Network (AN—IC2); Cerebellum (Cb—IC1); Default Mode Network (DMN—IC15, IC20, IC21, IC25, IC27, IC28, and IC31); Dorsal Attention System (DAS—IC6 and IC17); Fronto–Parietal Network (FPN—IC3 and IC23); Language Network (LN—IC 26); Salience Network (SN—IC 10 and IC19); Sensory Motor Network (SMN—IC7, IC8, and IC33) and Visual (IC11—Primary visual (PV) and IC 24—Higher visual (HVN)) Network.

### 4.4. Characterization of the BOLD RSNs by Higuchi’s Fractal Dimension

Higuchi fractal dimension is a measure of waveform complexity in the time domain [40]. A full discussion of the method with an explanation of the mathematical processes leading to the calculation of FD applied to the analysis of resting-state BOLD signal dynamics is given elsewhere [18,41,42].

#### Statistical Analysis

Firstly, we verified whether the IC’s FDs were normally distributed using the Kolmogorov–Smirnov test, and then we applied a general linear model considering groups and sex as factors, age as a covariate, and FDs as dependent variables.

Due to the ICs number (*n* = 9), the significance threshold was set at the *p*-value < 0.005 in order to correct for multiple comparisons.

We evaluated groups’ DTI metrics considering the following ROIs: whole hypothalamus anatomical area, anterior right and left hypothalamus, and posterior right and left hypothalamus.

All groups’ DTI metrics were normally distributed once verified by means of the Kolmogorov–Smirnov test. Descriptive statistics are shown in Table 1 and Table 2.

We applied general linear models considering groups and sex as factors, age as a covariate, and DTI metrics as dependent variables, for the whole hypothalamus and its 4 ROIs, respectively.

Cohen’s d was computed as a measure of effect size for FD and DTI metrics.

For DTI metrics inferential statistics, it was chosen a *p*-value of 0.01 to consider for multiple comparisons due to the number of hypothalamus ROIs (0.05/4 = 0.0125).

Pearson’s correlation coefficient was performed, respectively, between FD values for IC10 (number of correlations: 4, due to clinical variables) and DTI metrics values for hypothalamic ROI and clinical variables (for each ROI: 4 DTI metrics and 4 clinical variables = 16 correlations) collected from clinical files and a 1-month headache diary before the day of examination (the mean monthly severity of headache attacks, ranging from 0 to 10; the number of monthly migraine attacks; the mean monthly duration of the attacks, in hours; the monthly number of acute medications). The significance threshold was set at a *p*-value < 0.01.

## Figures and Tables

**Figure 1 ijms-23-13238-f001:**
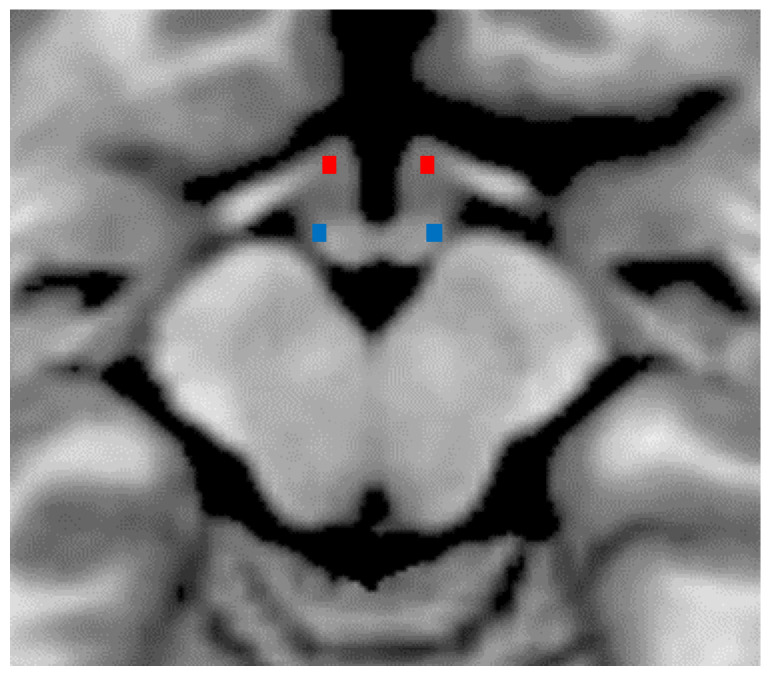
Bilateral anterior (in red) and posterior (in blue) hypothalamic regions of interest are shown on axial slice of an MRI template (T1-weighted MNI 2 mm).

**Figure 2 ijms-23-13238-f002:**
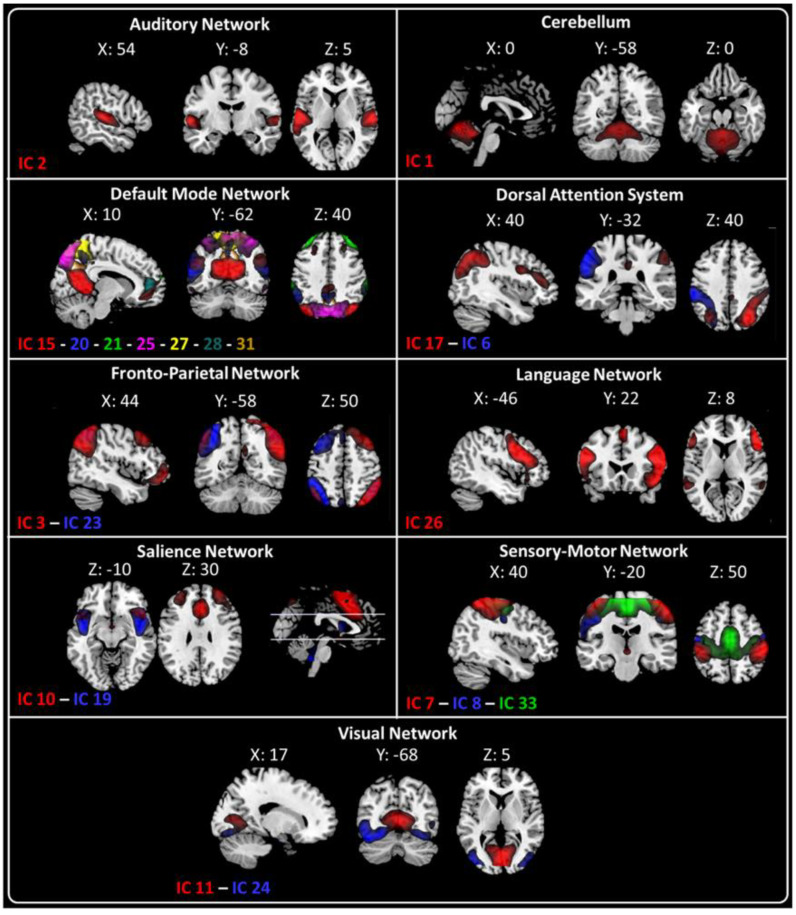
Twenty-one spatial maps divided into nine functional networks were found: Auditory (AN—IC2); Cerebellum (Cb—IC1); Default Mode (DMN—IC15, IC20, IC21, IC25, IC27, IC28, and IC31); Dorsal Attention System (DAS: IC6—lDAS and IC17—rDAS); Fronto–Parietal (FPN: IC3—rFPN and IC23—lFPN); Language (LN—IC 26); Salience (SN—IC 10 and IC19); Sensory Motor (SMN—IC7, IC8, IC33); and Visual (VN: IC11—Primary Visual and IC24—Higher Visual) networks based on their anatomical view. Montreal Neurological Institute (MNI) coordinates are shown as well.

**Figure 3 ijms-23-13238-f003:**
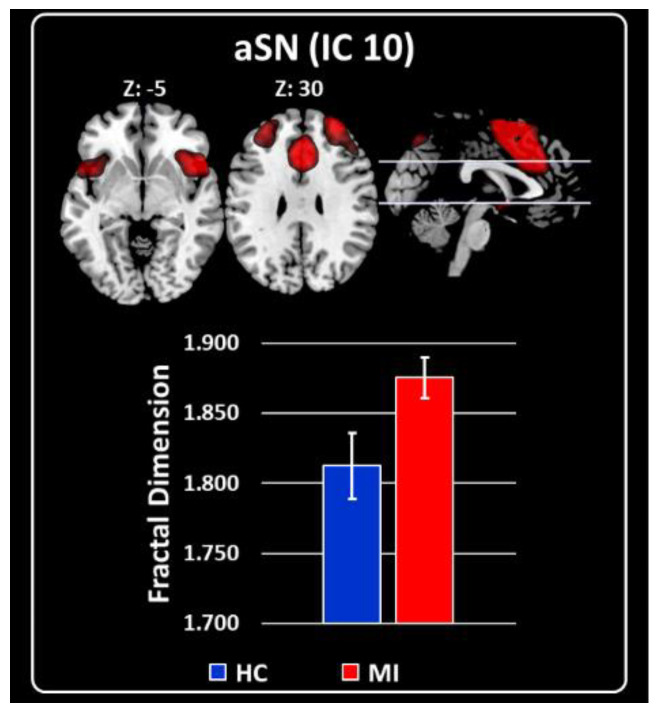
Grand average and standard error for the FD values are shown for both groups HC (blue) and MI (red). Upper panel shows the results obtained for aSN (IC 10). All were co-registered into the Montreal Neurological Institute (MNI) space.

**Figure 4 ijms-23-13238-f004:**
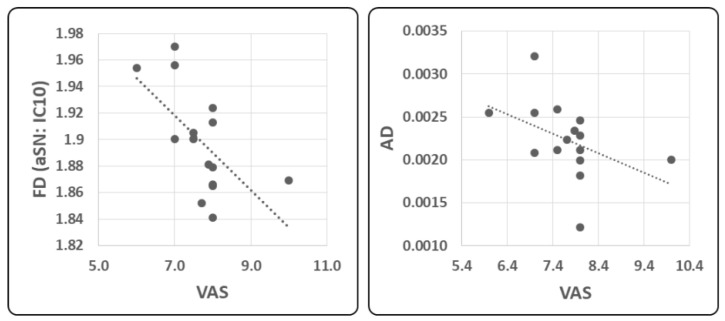
Linear regression analysis between FD values (dimensionless) of IC10 (aSN) BOLD activity and the mean severity of headache (VAS) (**left panel**) and between axial diffusivity (AD) DTI measure and VAS (**right panel**).

**Figure 5 ijms-23-13238-f005:**
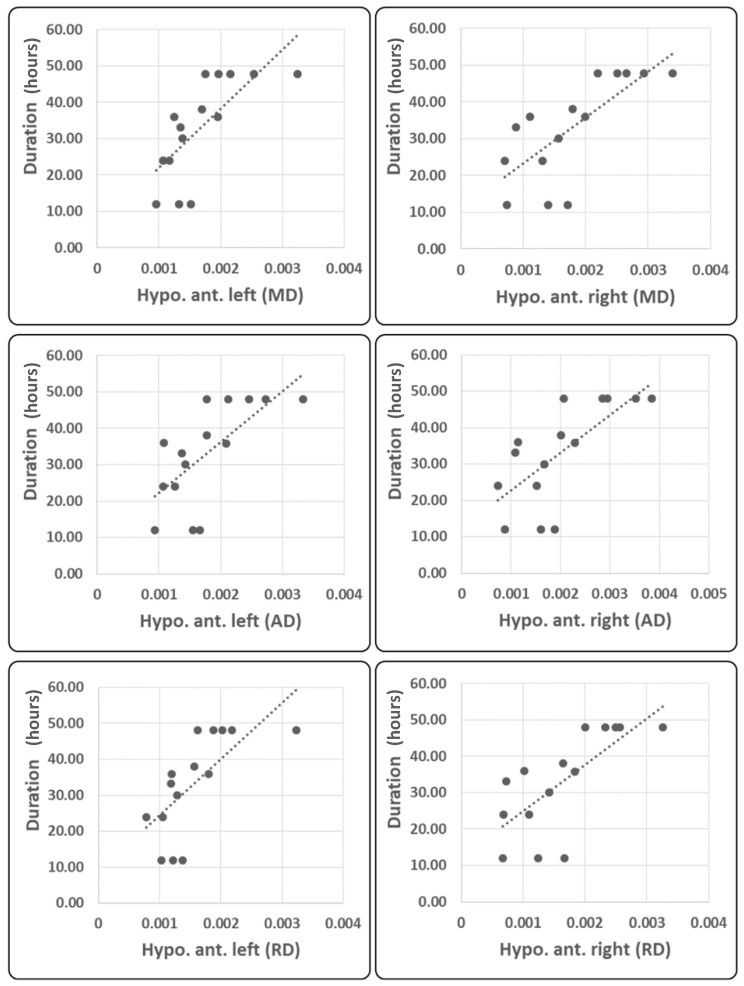
Linear regression analysis between duration of migraine attacks in hours and bilateral anterior hypothalamic MD, AD, and RD values.

**Table 1 ijms-23-13238-t001:** Clinical and demographic data of healthy controls (HC) and patients with migraine ictus (MI).

	Healthy Controls (HC: #20)	Ictal Migraine (MI: #15)
Women (number)	11	12
Age (years)	29.7 ± 3.94	34.3 ± 10.26
Attacks frequency/month (number)		4.5 ± 3.88
Mean attacks duration (h)		36.3 ± 30.0
Disease duration (year)		16.6 ± 9.33
Mean severity of headache (VAS) (0–10)		7.7 ± 0.86
Mean headache-related disability (number)		2.4 ± 0.43
Acute medication intake/month (number)		4.5 ± 3.32

Data are expressed as mean ± SD.

**Table 2 ijms-23-13238-t002:** Mean ± standard deviation for the hypothalamus fractional anisotropy (FA), mean diffusivity (MD), axial diffusivity (AD), and radial diffusivity (RD) in HC and MI. General linear models of DTI metrics inferences with the percentage of variation in the response that is explained by the model, adjusted for the number of predictors.

DTI Metric	HC	MI	Age (F; p)	Sex (F; p)	Group (F; p)	R2—adj (%)
**FA**	0.271 ± 0.050	0.149 ± 0.035	0.72; 0.404	0.19; 0.668	31.78; <0.001	62.11
**MD**	1.170E − 03 ± 1.630E − 04	2.157E − 03 ± 4.425E − 04	1.60; 0.217	1.08; 0.307	41.82; <0.001	69.64
**AD**	1.450E − 03 ± 1.530E − 04	2.238E − 03 ± 4.395E − 04	1.08; 0.307	1.48; 0.234	30.49; <0.001	62.02
**RD**	1.030E − 03± 1.690E − 04	2.021E − 03± 4.282E − 04	1.14; 0.295	1.37; 0.253	45.92; <0.001	73.77

**Table 3 ijms-23-13238-t003:** Mean ± standard deviation for the hypothalamus fractional anisotropy (FA), mean diffusivity (MD), axial diffusivity (AD), and radial diffusivity (RD) in HC and MI. General linear models of DTI metrics of each of the four ROIs of the hypothalamus with the percentage of variation in the response that is explained by the model, adjusted for the number of predictors.

Hypothalamic Region	DTI Metric	HC	MI	Age (F; p)	Sex (F; p)	Group (F; p)	R2—adj (%)
**Anterior Left**	**FA**	0.174 ± 0.054	0.159 ± 0.070	0.01; 0.930	0.36; 0.551	0.16; 0.694	0.00
	**MD**	1.520E − 03 ± 3.788E − 04	1.706E − 03 ± 6.391E − 04	1.35; 0.255	1.71; 0.202	0.17; 0.683	4.95
	**AD**	1.696E − 03 ± 4.514E − 04	1.806E − 03 ± 7.044E − 04	1.79; 0.192	0.71; 0.406	0.04; 0.848	0.36
	**RD**	1.433E − 03 ± 3.282E − 04	1.577E − 03 ± 6.369E − 04	2.38; 0.135	2.98; 0.096	0.03; 0.896	12.14
**Posterior Left**	**FA**	0.280 ± 0.161	0.116 ± 0.062	0.67; 0.421	1.30; 0.265	7.44; 0.01	28.25
	**MD**	1.576E − 03 ± 5.042E − 04	2.939E − 03 ± 4.739E − 04	0.21; 0.652	0.44; 0.513	35.94; <0.001	63.07
	**AD**	1.825E − 03 ± 4.961E − 04	3.190E − 03 ± 5.345E − 04	0.05; 0.822	0.16; 0.695	34.24; <0.001	61.03
	**RD**	1.468E − 03 ± 5.208E − 04	2.815E − 03 ± 4.070E − 04	0.02; 0.890	0.73; 0.402	41.33; <0.001	65.04
**Anterior Right**	**FA**	0.179 ± 0.065	0.190 ± 0.117	0.00; 0.946	0.03; 0.868	0.22; 0.643	0.00
	**MD**	1.361E − 03 ± 2.672E − 04	1.834E − 03 ± 8.640E − 04	5.24; 0.030	0.03; 0.870	0.44; 0.514	18.84
	**AD**	1.514E − 03 ± 2.860E − 04	2.046E − 03 ± 9.858E − 04	6.90; 0.014	0.01; 0.942	0.24; 0.625	22.41
	**RD**	1.281E − 03 ± 2.571E − 04	1.984E − 03 ± 8.173E − 04	4.62; 0.041	0.19; 0.668	0.33; 0.573	16.07
**Posterior Right**	**FA**	0.319 ± 0.133	0.117 ± 0.058	0.48; 0.493	0.55; 0.465	15.46; 0.001	43.32
	**MD**	1.754E − 03 ± 5.962E − 04	2.868E − 03 ± 4.083E − 04	0.79; 0.381	2.14; 0.156	21.82; <0.001	53.62
	**AD**	2.073E − 03 ± 5.808E − 04	3.145E − 03 ± 4.572E − 04	0.37; 0.550	1.57; 0.221	20.95; <0.001	50.54
	**RD**	1.608E − 03 ± 6.013E − 04	2.795E − 03 ± 3.166E − 04	0.15; 0.703	2.46; 0.129	30.48; <0.001	58.82

## Data Availability

The data that support the findings of this study are available from the corresponding author upon reasonable request.

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
