# Peer review of "A Hypothalamic Mechanism Regulates the Duration of a Migraine Attack: Insights from Microstructural and Temporal Complexity of Cortical Functional Networks Analysis"

_ijms, 2022, doi:10.3390/ijms232113238_

Round 1

Reviewer 1 Report

This manuscript applied DTI and fMRI to reveal the hypothalamic mechanism during a migraine attack. While the topic seems to be interesting, I have strong concerns about the method part. I hope the authors could provide more details. Here are my comments:

1. The imaging parameters are largely unknown, especially for the resolution. This is very important to take into consideration when performing image normalization/registration.

2. DTI preprocessing: Not sure if the analysis was performed at subject space or template space. And the ROI size is very very tiny. If it is performed at subject space, how to ensure the ROI location is accurate. If it is performed at template space, what kind of procedure was performed to align DTI image to template? And, please also provide the coornidates of each ROI.

3. fMRI preprocessing missed lots of common steps, which would cause strong noise in the BOLD signal. A common pipeline should includes: removing first 5-10 volumes, slice-time correction, motion correction, bandpass filtering, regressing WM/CSF/Global signal, and regressing motion parameters.

4. T1 normalization procedure is also unclear. What type of algorithm was used? This is very important and would influence the registration accuracy, since this paper used very tiny ROI.

5. What is the criteria to select 50 or 33 ICA components? I think there are multiple algorithms could automatically estimate the number of components. And, the combination process of RSN networks needs to be explored more. Please provide the spatial correlation values in the supporting material. Otherwise, these components seem to be a manually selected.

Author Response

This manuscript applied DTI and fMRI to reveal the hypothalamic mechanism during a migraine attack. While the topic seems to be interesting, I have strong concerns about the method part. I hope the authors could provide more details. Here are my comments:

  1. The imaging parameters are largely unknown, especially for the resolution. This is very important to take into consideration when performing image normalization/registration.

Answer: MRI data were obtained on a Siemens 3T Verio scanner using a 12-channel head coil. The parameters of each structural and functional sequences are now added in the main body text, as follows:

Paragraph 4.2:

“Diffusion tensor imaging (DTI) was acquired by using single shot echo-planar imaging, with an 12–channel head coil (TR 12200 ms, TE 94 ms, 72 axial slices, 2 mm thickness, isotropic voxels) [18], using SPAIR (Spectral Attenuated Inversion Recovery) as fat suppression technique.

Images from the same participants and during the same session were obtained with diffusion gradients applied along 30 non-collinear directions, effective b values of 0 and 1000 s/mm2 were used.”

Paragraph 4.3:

“MRI data were obtained on a Siemens 3T Verio scanner using a 12-channel head coil.

Structural anatomic scans were performed using a T1-weighted sagittal magnetization-prepared rapid gradient echo (MPRAGE) series (TR: 1900ms, TE: 2.93ms, 176 sagittal slices, 0.508 x 0.508 x 1 mm3 voxels).

Functional MRI resting state data were obtained using T2*-weighted, echo-planar imaging (TR: 3000ms, TE: 30ms, 40 axial slices, 3.906 x 3.906 x 3 mm3, 150 volumes).”

  1. DTI preprocessing: Not sure if the analysis was performed at subject space or template space. And the ROI size is very very tiny. If it is performed at subject space, how to ensure the ROI location is accurate. If it is performed at template space, what kind of procedure was performed to align DTI image to template? And, please also provide the coordinates of each ROI.

Answer: We would like to thank the Reviewer for giving us the opportunity to clarify this important point. We recognise that the ROIs are very small, but they are exactly the same as those published by Boes and colleagues (2018) [1] and in other articles by the research group of Michael D. Fox. We also agree that the smaller the area of interest, the larger the variability might be. However, we must note that, from a quick analysis of studies that have analyzed the hypothalamus in migraine patients, the sizes of our ROIs are in some cases larger than those previously used, and consequently less prone to variability (see table below). We have already considered this point as a limitation of the present study (lines 270-273).

Authors

Coordinates

ROI

Denuelle et al., (Denuelle et al., 2007)

RH: x = 2, y = −8, z = −10

LH: x = −4, y = 4, z = −12

Schulte et al., (Schulte and May, 2016)

RH: x = 0, y = 2, z = -6

Moulton et al., (Moulton et al., 2014)

x = -6, y = -2, z = -8

8 mm3 sphere

Schulte et al., (Laura H Schulte et al., 2020)

x = -6, y = -2, z = -8

8 mm3 sphere

Schulte et al., (Laura H. Schulte et al., 2020)

RH: x = 0, y = 2, z = -6

6 mm3 sphere

Meylakh et al., (Meylakh et al., 2020)

X = 14, y = -4, z = -10

Present article

Anterior: x = ±4.4, y = 1.33, z = -14.67

Posterior: x = ±6.6, y = -7.33, z = -12.67

48 mm3 parallelepiped  per hemisphere

Two expert neuroradiologists (E.T. & F.C.) drew each ROI on the template MNI152_T1_2mm_brain.nii.gz provided in FSL, based on Baroncini et al. [2] and Boes et al. [1] methods.

The center of gravity coordinates in MNI space for the anterior hypothalamic region are x = ±4.4; y = 1.33; z = −14.67, the posterior hypothalamic ROI coordinates are x = ±6.6; y = −7.33; z = −12.67 and those of whole hypothalamic ROI are x = 0.136; y = 4.68; z = -11.9.

The size of the ROIs in 2 mm space was 6 voxels (48 mm3), 3 voxels per hemisphere, and was the same for both anterior and posterior hypothalamus ROIs.

Each participants’ DTI metrics (AD, FA, MD and RD) were registered with the template cited before as reference image, using a rigid body model (6 parameters) and mutual information as Cost Function.

The participants' anatomical T1 scans were not used in any of the DTI processing steps.

All this information has now been added to the section 4.2 on DTI data acquisition and analysis.

  1. fMRI preprocessing missed lots of common steps, which would cause strong noise in the BOLD signal. A common pipeline should includes: removing first 5-10 volumes, slice-time correction, motion correction, bandpass filtering, regressing WM/CSF/Global signal, and regressing motion parameters.

Answer: We discarded the first 5 volume of each participant and this sentence is now included in the main body text (added at lines 362-363).

Slice-time correction was not necessary due to the repetition time (TR = 3 sec) of our scanner.

Motion correction was performed as already stated in the main body text (lines 364-370).

The independent component analysis (ICA) method is able to detect resting state fluctuations from other noise effects [3,4].

Gift software does not involve regressors to analyse pre-processed fMRI data (see recommendations in https://trendscenter.org/software/gift/).

  1. T1 normalization procedure is also unclear. What type of algorithm was used? This is very important and would influence the registration accuracy, since this paper used very tiny ROI.

Answer: We directly registered each participant’s DTI metrics with the template MNI152_T1_2mm_brain, without normalizing single participant’s T1 structural image to the template as stated in the answer to point 2 of this Response-to-Reviewer letter. Each participant’s T1 scan and realigned EPI-BOLD images were normalized into a common stereotactic space (see lines 372-374).

The normalization procedure was performed based on TPM.nii as tissue probability maps, affine registered with ICBM space template (European brains) and interpolated with a 4th degree b-spline.

  1. What is the criteria to select 50 or 33 ICA components? I think there are multiple algorithms could automatically estimate the number of components. And, the combination process of RSN networks needs to be explored more. Please provide the spatial correlation values in the supporting material. Otherwise, these components seem to be a manually selected.

Answer: As stated in the main body text, we propose using standard information-theoretic methods for estimating the number of components from the aggregate data set. These methods make a decision based upon the complexity or information content of the data. The number of ICs was estimated using the minimum description length (MDL) criterion (see lines from 389 to 394 and the reference articles 35 & 36).

In our specific case, 33 independent components (ICs) were the defined by the above analysis. Subject-specific spatial maps and time courses were obtained using the back-reconstruction approach (GICA) (lines 391 e 393, reference 37).

We did not perform an automatic classification of meaningful ICs, but this procedure was manual, based on well-known reference articles (34, 38-39). Two experienced neuroradiologists blindly recognized meaningful ICs and sorted into functional networks [5]. We have modified the manuscript accordingly (line 397-401).

References

  1. Boes, A.D.; Fischer, D.; Geerling, J.C.; Bruss, J.; Saper, C.B.; Fox, M.D. Connectivity of sleep- and wake-promoting regions of the human hypothalamus observed during resting wakefulness. Sleep 2018, 41, zsy108, doi:10.1093/sleep/zsy108.
  2. Baroncini, M.; Jissendi, P.; Balland, E.; Besson, P.; Pruvo, J.-P.; Francke, J.-P.; Dewailly, D.; Blond, S.; Prevot, V. MRI atlas of the human hypothalamus. Neuroimage 2012, 59, 168–180, doi:10.1016/j.neuroimage.2011.07.013.
  3. Beckmann, C.; DeLuca, M.; Devlin, J.T.; Smith, S.M. Investigations into resting-state connectivity using independent component analysis. Philos. Trans. R. Soc. LondonSeries B, Biol. Sci. 2005, 360, 1001–1013.
  4. Jafri, M.; Pearlson, G.D.; Stevens, M.; Calhoun, V.D. A method for functional network connectivity among spatially independent resting-state components in schizophrenia. Neuroimage 2008, 39, 1666–1681.
  5. Griffanti, L.; Douaud, G.; Bijsterbosch, J.; Evangelisti, S.; Alfaro-Almagro, F.; Glasser, M.F.; Duff, E.P.; Fitzgibbon, S.; Westphal, R.; Carone, D.; et al. Hand classification of fMRI ICA noise components. Neuroimage 2017, 154, 188–205, doi:10.1016/j.neuroimage.2016.12.036.

Reviewer 2 Report

The current manuscript conducted DTI and functional MRI analysis among 15 patients with migraine and 20 healthy controls. Based on the parameters, the authors discussed the correlation between the hypothalamus microstructure and salience network with migraine attacks. The introduction and methods parts are well-written and provided sufficient background and details to the readers. The discussion part provided in-depth thinking and also pointed out the limitation of the current study.  I have a few comments below:

1. If I understand correctly, all the participants are females in this study. Is there a sex difference in the current findings?

2. What is the rationale for choosing the early afternoon for scanning? Since the hypothalamus is where the master circadian clock is located, does the circadian clock also affect the severity of migraine attacks?

3. Please explain each abbreviation the first time it occurs. For example, in line 144, please explain "GLM" and aSN; line 160, BOLD.

4. The resolution of Figure 1 is too low. Is it possible to replace it with a clearer image?

5. The figure legend in Figure 4 is difficult to understand. What does the y-axis of both panels represent? Please rephrase the legend.

6. In Figure 5, what is the unit of Duration? 

7. The results in the current manuscript are not enough to make a conclusion that "the complexity of the SN involved in the multidimensional  neurocognitive processing of pain". Any further evidence to support it?

8. Some sentences in the manuscript are a bit confusing. It would be helpful if the authors can have an English native speaker read and edit the manuscript. 

Author Response

The current manuscript conducted DTI and functional MRI analysis among 15 patients with migraine and 20 healthy controls. Based on the parameters, the authors discussed the correlation between the hypothalamus microstructure and salience network with migraine attacks. The introduction and methods parts are well-written and provided sufficient background and details to the readers. The discussion part provided in-depth thinking and also pointed out the limitation of the current study.  I have a few comments below:

  1. If I understand correctly, all the participants are females in this study. Is there a sex difference in the current findings?

Answer: We thank the Reviewer for giving us the opportunity to clarify this point. As stated in Table 1, 11 out of 20 healthy controls were women, while 12 out of 15 were ictal migraineurs. Our statistician advised us that the number of male participants in the patient group is too small to be able to proceed with a statistical analysis of the differences.

  1. What is the rationale for choosing the early afternoon for scanning? Since the hypothalamus is where the master circadian clock is located, does the circadian clock also affect the severity of migraine attacks?

Answer: The choice was limited by availability of the MRI machine which is shared with other research groups leaving ours with an afternoon time window. However, we think that this limitation is mitigated by the fact that both groups (controls and patients) were recorded in the afternoon. We have added this as an area of future investigation as follows: “Further studies designed to evaluate circadian variability of hypothalamic measures might shed light on the possible sensitivity of diffusion parameters to the different phases of the day and night.” (lines 285-288).

  1. Please explain each abbreviation the first time it occurs. For example, in line 144, please explain "GLM" and aSN; line 160, BOLD.

Answer: added, thank you.

  1. The resolution of Figure 1 is too low. Is it possible to replace it with a clearer image?

Answer: Now, we have replaced Figure 1 with a more resolute one, and we have changed the Figure Legend as follows: “Bilateral anterior (in red) and posterior (in blue) hypothalamic regions of interest are shown on axial slice of an MRI template (T1-weighted MNI 2 mm).”

  1. The figure legend in Figure 4 is difficult to understand. What does the y-axis of both panels represent? Please rephrase the legend.

Answer: We accept the reviewer’s point. We corrected the figure legend as follows: “Linear regression analysis between FD values (dimensionless) of IC10 (aSN) BOLD activity and the mean severity of headache (VAS) (left panel), between axial diffusivity (AD) DTI measure and VAS (right panel)”.

  1. In Figure 5, what is the unit of Duration?

Answer: Hours, now added.

  1. The results in the current manuscript are not enough to make a conclusion that "the complexity of the SN involved in the multidimensional neurocognitive processing of pain". Any further evidence to support it?

Answer: The Reviewer is right. Now, we have re-phrased the pinpointed sentence of the abstract as follows: “….the functional connectivity of the SN involved in the multidimensional neurocognitive processing of pain.”.

  1. Some sentences in the manuscript are a bit confusing. It would be helpful if the authors can have an English native speaker read and edit the manuscript.

Answer: Now, the text has been proofread and edited by native English-speaking colleague.

Round 2

Reviewer 1 Report

I appreciated the author's clarification of all the comments. I still have concerns about the data preprocessing, for instance, the accuracy of the registration, and the lack of slice time correction for 3s TR. 3s is a long time for TR, I don't see the reason to ignore the slice time correction. If the TR is 0.8s, I think it is ok to ignore this step. Meanwhile, manually combining ICs into networks is not a solid solution. I previously suggested the authors to provide the inter-IC correlation, which may help explain the rationale of the combination. However, no results were provided.

Author Response

I appreciated the author's clarification of all the comments. I still have concerns about the data preprocessing, for instance, the accuracy of the registration, and the lack of slice time correction for 3s TR. 3s is a long time for TR, I don't see the reason to ignore the slice time correction. If the TR is 0.8s, I think it is ok to ignore this step. Meanwhile, manually combining ICs into networks is not a solid solution. I previously suggested the authors to provide the inter-IC correlation, which may help explain the rationale of the combination. However, no results were provided.

Answer: We did not perform slice time correction. Slice-timing correction is not mandatory because the haemodynamic response is longer than TR (about 3.0 s). Moreover, the EPI sequence we have used is a multiband sequence with simultaneous echo refocusing and parallel imaging (called GRAPPA by Siemens), that makes slice time correction problematic and vulnerable to the slightest movement. For these reasons we chose to skip this habitual step in fMRI analysis.

As we have already stated in the previous revision, Gift software contains only the maps of the DMN and bilateral visuomotor network.

However, the software tool includes a file called “RSN.nii” in which most common networks’ ROIs are embedded, therefore now we extracted each network using an FSL’s procedure, following the specified list in the RSN.txt file. Lastly, we calculated correlation’s values for each network previously identified with reference to the networks’ templates, and we prepared the following table, now added as supplementary material (lines 361-362):

Network

Independent Component, correlation index

Cerebellum

IC1

0.36

Salience

IC10, IC19

0.32, 0.25

Auditory

IC2

0.38

Dorsal attention

IC6, IC17

0.21, 0.24

Default Mode

IC27, IC28, IC31

0.10, 0.64, 0.19

 IC25, IC21

 0.17, 0.17

IC20, IC15

0.29, 0.25

Frontal parietal

IC3, IC23

0.31, 0.32

Language

IC26

0.21

Sensory motor

IC33, IC7

0.21, 0.19

IC8

0.14

Visual

IC24

0.36

IC11

0.22

Supplementary Table 1: Correlation’s values for each network previously identified with reference to the networks’ templates

Reviewer 2 Report

The authors resolved my concerns and made great improvements in the revised manuscript. I accept the changes in the revision.

Author Response

The authors resolved my concerns and made great improvements in the revised manuscript. I accept the changes in the revision.

Answer: We thank the Reviewer for his/her positive comment.

Round 3

Reviewer 1 Report

I disagree with the author's statement about slice time correction. 3s is a long TR compared with the currently used 0.8s. Multiband slice time correction is also doable. I don't think this process is questionable. The editor could judge this point. 

And I don't have other comments.